# From Menopause to Molecular Dysregulation: Proteomic Insights into Obesity-Related Pathways—A Narrative Review

**DOI:** 10.3390/biomedicines13071558

**Published:** 2025-06-25

**Authors:** Basant E. Katamesh, Jithinraj Edakkanambeth Varayil, Nina Pillai, Ann Vincent

**Affiliations:** 1Division of General Internal Medicine, Mayo Clinic, Rochester, MN 55905, USA; el-fetouhkatamesh.basant@mayo.edu; 2Department of Family Medicine, Mayo Clinic, Rochester, MN 55905, USA; edakkanambethvarayil.jithinraj@mayo.edu; 3Edward Via College of Osteopathic Medicine, Virginia Campus (VCOM–VA), Blacksburg, VA 24060, USA

**Keywords:** menopause, overweight, obesity, proteomics

## Abstract

Peri- and postmenopausal women often experience unexplained weight gain despite maintaining consistent dietary and lifestyle habits. While the biological mechanisms underlying this phenomenon remain poorly understood, physiological and pathophysiological changes during the menopausal transition are likely contributors. Proteomic profiling holds potential for revealing key molecular pathways involved in the pathogenesis of obesity in this population. This review synthesizes current evidence on proteomic alterations linked to overweight and obesity in peri- and postmenopausal women. A structured literature search was performed across Ovid MEDLINE^®^, EMBASE, the Cochrane Library, and Scopus for studies published between October 2010 and March 2025. Eligible studies included original research involving overweight or obese peri- or postmenopausal women that reported proteomic data. Extracted information encompassed study design, participant characteristics, sample types, and proteomic findings. Identified proteins were cross-referenced with a prior review of consistently dysregulated proteins in obesity. Five studies met the inclusion criteria, collectively revealing consistent proteomic patterns associated with inflammation, metabolic dysfunction, and endothelial dysregulation. These included C-reactive protein, Tissue necrotic factor-alpha, interleukins, adiponectin, and endocan. Notably, one study demonstrated that weight loss led to reductions in IL-6, IL-1 receptor antagonist, and CRP, suggesting that obesity-related inflammation may be at least partially reversible. This review provides preliminary evidence linking chronic inflammation, metabolic dysregulation, and vascular stress to obesity in peri- and postmenopausal women. These proteomic signatures enhance understanding of menopausal weight gain and highlight the potential of proteomics to guide personalized interventions. However, larger, well-designed prospective studies are needed to confirm these associations and clarify causal pathways.

## 1. Introduction

In clinical practice, it is not uncommon to encounter otherwise healthy peri- and postmenopausal women who experience a perplexing pattern of unexplained weight gain, typically between 5 and 15 pounds, despite maintaining disciplined dietary and lifestyle habits. The exact etiology of this weight gain remains incompletely understood. However, epidemiological studies support this clinical observation, indicating that the menopausal transition is associated with increased total body weight, central obesity, and visceral adiposity [1,2,3]. For example, the Study of Women’s Health Across the Nation (SWAN), which followed 1246 women with a mean age of 47.1 years, found that fat mass and BMI began to rise approximately 5–6 years before menopause and continued to increase for up to four years afterward [4]. Similarly, the Nurses' Health Study (NHS) reported an average weight gain of 6.8 pounds over an 8-year period among women transitioning through menopause, equating to roughly 0.85 pounds per year [5].

These clinical and epidemiological findings underscore the need to better understand the physiological mechanisms driving this weight gain. Hormonal and metabolic changes during the menopausal transition are thought to play a central role [6,7]. The decline in estrogen levels alters lipid and fatty acid metabolism, promoting visceral fat accumulation and a shift in fat distribution from a gynoid (hip/thigh) to an android (abdominal) pattern [8,9,10]. In premenopausal women, adipose tissue expansion primarily occurs via hyperplasia, favoring subcutaneous fat deposition, which is associated with a lower risk of metabolic disease [11,12]. In contrast, postmenopausal fat accumulation is characterized by a hypertrophic expansion of the visceral adipose tissue (VAT), which is more metabolically active and pro-inflammatory than subcutaneous fat, thereby increasing cardiometabolic risk [13].

In the general population, obesity is linked to elevated oxidative stress and inflammation via the release of pro-inflammatory adipokines (such as leptin, visfatin, and resistin), anti-inflammatory adipokines (such as adiponectin and apelin), chemotactic factors (such as MCP-1), and pro-inflammatory cytokines (such as TNF-α, IL-1β, and IL-6) [14,15]. Whether obesity associated with the menopausal transition elicits a similar inflammatory and oxidative stress profile remains unclear and warrants further investigation [16]. Clarifying this relationship is critical, as these pathophysiological mechanisms are central to the development of numerous chronic diseases, including cardiovascular disease, hypertension, dyslipidemia, non-alcoholic fatty liver disease, osteoarthritis, and certain cancers [17]. As obesity becomes increasingly common and clinically significant during the menopausal transition, understanding the underlying biochemical and physiological drivers of adiposity and its sequelae is of critical importance.

One effective approach to uncovering these mechanisms is through proteomics, the large-scale study of proteins and their functions. Proteomics offers a promising avenue for identifying the molecular mechanisms that drive obesity and its related complications, including metabolic dysfunction, inflammation, and cellular stress [18]. By analyzing patterns of protein expression and regulation, proteomics can provide a system-level understanding of the biochemical disruptions associated with excess adiposity [18]. One systematic review has demonstrated this potential by synthesizing proteomic alterations linked to obesity [18]. The review included 301 participants (149 normal-weight and 152 obese individuals, aged 24 to 52 years) and analyzed a wide range of biological samples, including extracellular vesicles, platelets, subcutaneous and visceral adipose tissue, skeletal muscle, endometrial tissue, granulosa cells, and sperm [18]. Across these studies, 362 proteins were found to be significantly dysregulated between obese and normal-weight individuals [18]. Of these, 41 proteins were consistently reported in at least two studies, suggesting potential biological relevance [18]. These proteins were involved in key pathways such as metabolism (14 proteins), inflammation and oxidative stress (7), cytoskeletal structure (7), coagulation (7), and chaperone binding (6), reflecting the multifactorial nature of obesity pathophysiology [18]. Notably, several of these proteins have also been implicated in obesity-related conditions, such as type 2 diabetes, non-alcoholic fatty liver disease, cardiovascular disease, and metabolic syndrome [18].

Despite these insights, few studies have focused specifically on postmenopausal women, who experience unique hormonal and metabolic changes. This gap underscores the need to characterize proteomic signatures of obesity in this population, a goal our study aims to address.

## 2. Materials and Methods

A structured literature search was conducted across Ovid MEDLINE®, EMBASE, Cochrane Library, and Scopus to identify studies published between October 2010 and March 2025. Search terms included “menopause”, “perimenopause”, “postmenopause”, “proteomics”, “glycoproteomic” and related keywords and MeSH terms.

Studies were eligible for inclusion if they were original research articles, either observational or interventional in design, and investigated proteomic markers and/or pathways in peri- or postmenopausal women with overweight or obesity. Studies were excluded if they lacked proteomic data, involved only premenopausal participants, animal, or cellular studies, or were reviews, editorials, or non-English articles.

Titles and abstracts were screened, followed by a full-text review of eligible studies. The data extracted included study design, population characteristics, sample type, and key proteomic findings associated with overweight/obesity, weight change, and metabolic alterations. Supplementary files were also reviewed to identify additional proteomic markers not reported in the main text or tables. Specifically, we cross-referenced these materials with a list of 41 proteins identified in a prior systematic review as consistently reported across at least two studies [18].

## 3. Results

Five studies met the inclusion criteria, including four cohort and one cross-sectional study (Table 1) [19,20,21,22,23]. Studies were conducted in the United States (3), Brazil (1), and Montenegro (1), with sample sizes ranging from 70 to 1534 postmenopausal women [19,20,21,22,23]. All studies focused on overweight/obesity-related biomarkers in postmenopausal women using serum, plasma, or whole blood samples [19,20,21,22,23].

Two studies examined proteomic markers associated with metabolic syndrome in postmenopausal women, both reporting pro-inflammatory shifts in this demographic [20,21]. Sinatora et al. (2022) conducted a cross-sectional study involving 70 obese postmenopausal women to examine the relationship between inflammatory markers and metabolic syndrome [21]. They reported significantly lower TNF-α levels and higher IL-10/TNF-α ratios among participants with metabolic syndrome (*p* = 0.005), suggesting a pro-inflammatory state [21]. Similarly, Klisic et al. (2023) examined the association of serum endocan with metabolic syndrome in a cohort of 126 postmenopausal women [20]. They found significantly higher serum endocan levels (*p* < 0.001) and increased C-reactive protein levels (*p* < 0.001) in the metabolic syndrome group, indicating higher levels of endothelial dysfunction and systemic inflammation [20].

The three remaining studies explored broader associations between obesity and metabolic, inflammatory, or proteomic biomarkers [19,22,23]. Stevens et al. (2020) analyzed metabolomic profiles in 1534 postmenopausal women and found that higher BMI and waist circumference were significantly associated with elevated serum CRP and C-peptide levels, while adiponectin levels were inversely associated (all *p* < 0.001), underscoring the link between adiposity and systemic inflammation [19].

Similar findings were reported by Garrison et al. (2017), who conducted a proteomic analysis on 924 plasma samples from postmenopausal women, identifying significant dysregulation in inflammatory, cellular stress, and metabolic pathways in overweight/obese individuals compared to those with normal BMI [19]. Additionally, cross-referencing their supplementary data with the 41 proteins identified by Rodriguez-Muñoz et al. uncovered nine additional obesity-associated proteins not reported in the main text [18,19].

In the fifth study, Wong et al. (2008) analyzed blood cytokine profiles in 290 overweight or obese postmenopausal women and found that weight loss was associated with significant reductions in IL-1 receptor antagonist, IL-6, and C-reactive protein levels (*p* < 0.05), indicating a favorable shift in inflammatory status following lifestyle intervention [23].

## 4. Discussion

To the best of our knowledge, this is the first narrative review to comprehensively compile and evaluate proteomic findings specific to overweight and obese postmenopausal women. By integrating data across existing studies, it provides valuable insights into molecular alterations that may influence disease development and inform the design of targeted therapeutic strategies.

Proteomic profiles of all included studies consistently demonstrate an upregulation of inflammatory biomarkers in overweight and obese postmenopausal women [19,20,21,22,23]. Elevated levels of IL-6, TNF-alpha, CRP, MCP-1, and endothelial dysfunction biomarkers such as oxidized HDL and soluble ICAM-1 supports the role of inflammation in the development and persistence of visceral adiposity and systemic inflammation in this population [24,25,26,27]. These findings are further supported by Wong et al., who reported that weight loss in postmenopausal women produced meaningful alterations in inflammatory markers such as IL-6, CRP, IL-1 receptor antagonist, IL-10, and TNF-α [23]. Additionally, a prospective longitudinal study tracking 69 women from pre- to post-menopause, reported that increases in visceral fat during the menopausal transition were associated with inflammatory markers, such as CRP, leptin, serum amyloid A, and tissue plasminogen activator antigen, and inversely associated with adiponectin [28]. These observations align with our study, which identified proteomic biomarkers indicative of heightened cellular stress and inflammatory activity in overweight/obese postmenopausal women. Collectively, these findings suggest that the menopausal transition may trigger a pathophysiological shift toward a pro-inflammatory state [19,20,21,22,23,28]. However, it remains unclear whether this response is universal among all women or limited to specific subgroups.

In addition to markers of chronic inflammation, two of the five studies revealed proteomic signatures suggestive of metabolic dysregulation [19,22]. These studies reported significant alterations in critical metabolic pathways, including insulin signaling, PI3K–Akt, and FoxO signaling—pathways essential for regulating glucose metabolism, lipid processing, and energy balance [19,22]. Additionally, Garrison et al. (2017) identified disruptions in insulin receptor signaling and hormonal response pathways, along with abnormalities in AGE–RAGE signaling and Oncostatin M activity, highlighting the interplay between inflammation and metabolic dysfunction associated with excess adiposity [19]. Similarly, Stevens et al. (2020) reported elevated C-peptide levels and reduced adiponectin concentrations, markers indicative of impaired insulin sensitivity, and increased cardiometabolic risk in women with higher BMI and waist circumference [22]. Together, these findings highlight the complex interplay between inflammation and metabolic dysregulation in postmenopausal women with obesity, suggesting that both processes may act synergistically to exacerbate health risks during the menopausal transition [19,22].

Another notable finding was the identification of proteomic patterns consistent with endothelial dysfunction in two of the five studies, alongside evidence of chronic inflammation [20,21]. Both studies observed elevated markers of endothelial activation and vascular inflammation in overweight and obese postmenopausal women [20,21]. Sinatora et al. (2022) reported altered inflammatory cytokine ratios, notably a significantly increased IL-10/TNF-α ratio, indicating immune imbalance and endothelial stress, while Klisic et al. (2023) observed significantly elevated circulating levels of endocan and C-reactive protein (CRP) in women with metabolic syndrome, consistent with subclinical endothelial dysfunction [20,21]. These findings highlight the potential vascular consequences of excess adiposity in postmenopausal women and suggest that endothelial impairment may further contribute to systemic inflammation.

Findings from the reviewed proteomic studies suggest a common biological signature marked by inflammation, metabolic disturbances, and endothelial dysfunction in overweight and obese women following menopause [19,20,21,22,23]. These processes may act independently or interactively, compounding one another through complex biochemical pathways that are particularly active during and after the menopausal transition [21,29,30]. While individual proteomic markers reflect distinct disruptions in immune function, insulin signaling, and vascular health, their convergence points to a broader, integrated pathophysiologic state specific to this high-risk population.

The estrogen deficiency accompanying menopause is a likely contributor to the physiological and molecular disruptions observed in this population [31,32]. Estrogen plays a vital regulatory role across numerous physiological systems, and its loss can lead to widespread systemic changes [33]. In overweight and obese postmenopausal women, this hormonal shift is further complicated by increased visceral adiposity and aromatase activity in visceral adipose tissue, which produces estrone (E1) [31,32]. However, estrone does not replicate the protective effects of ovarian estradiol [34]. In the context of chronic low-grade inflammation and insulin resistance, both hallmarks of obesity, estrone may lose its beneficial properties and potentially contribute to pathological processes [31,32].

In premenopausal women, ovarian estradiol helps regulate immune function by suppressing the NF-κB pathway, enhancing anti-inflammatory cytokines such as IL-10, and promoting macrophage polarization toward the anti-inflammatory M2 phenotype [35,36,37]. Nonetheless, the inflammatory proteomic patterns reported across the five studies imply that, in overweight and obese postmenopausal women, existing estrogen levels may be functionally inadequate or unable to counteract the heightened pro-inflammatory environment associated with excess adiposity. It is also plausible that the decline of other ovarian hormones during the peri- and postmenopausal periods could contribute to these changes. Since none of the included studies directly measured estrogen levels or other hormone levels in participants, this hypothesis remains speculative and requires further investigation. 

This review has several strengths and limitations. It offers a targeted synthesis of proteomic alterations linked to overweight and obesity in postmenopausal women, a population at heightened cardiometabolic risk due to hormonal and physiological transitions. A major strength of this work lies in its rigorous and comprehensive literature search strategy, which utilizes multiple major databases to ensure broad coverage. Furthermore, cross-referencing findings with a previously published systematic review allowed for the inclusion of all known obesity-associated proteomic markers, thereby enhancing the biological plausibility and relevance of the identified pathways.

Nonetheless, several limitations should be acknowledged. The number of eligible studies was limited, and sample sizes varied considerably, reducing the statistical power and generalizability of the findings. Additionally, all included studies employed observational designs, which preclude causal inference. Considerable heterogeneity in the proteomic targets assessed, with minimal overlap in specific proteins or pathways, and varying cytokine and proteomic panels selected by investigators due to the lack of a standardized profile for obesity in menopause further limited direct comparisons and the identification of consistent biomarkers. Additionally, as a narrative rather than a systematic review, this study did not formally assess bias or study quality and may lack the methodological reproducibility of systematic reviews. Despite these constraints, this review highlights key proteomic patterns relevant to menopause-associated obesity and provides a foundation for future research. Continued investigation is essential to determine the diagnostic and therapeutic potential of these biomarkers and to guide targeted interventions aimed at reducing visceral adiposity and systemic inflammation in postmenopausal women.

## 5. Conclusions

This review highlights the complex and multifaceted biological mechanisms underlying obesity in postmenopausal women, emphasizing consistent proteomic alterations across inflammatory, metabolic, and cellular stress pathways. These molecular signatures point to a state of systemic dysregulation that is not fully captured by traditional anthropometric measures and underscore the unique metabolic challenges posed by the menopausal transition. By synthesizing current proteomic evidence specific to this high-risk population, this study offers valuable insights into the pathophysiology of menopause-associated adiposity and its broader metabolic implications.

As proteomic technologies continue to evolve, future research should focus on validating these biomarkers in larger, longitudinal cohorts and investigating their potential to guide early, personalized interventions. Ultimately, a deeper understanding of these molecular pathways could inform more effective strategies to mitigate visceral adiposity and systemic inflammation in postmenopausal women.

## Figures and Tables

**Table 1 biomedicines-13-01558-t001:** Summary of proteomics studies.

Author, Year	- Study Design - Number of Participants - Sample Tested - Country	Aim of the Study	Inclusion Criteria	Study Findings
Sinatora et al., 2022 [1]	- Cross-sectional study - 70 participants - Serum samples - Brazil	Relationship between inflammatory markers and metabolic syndrome in postmenopausal women	(1) Postmenopausal women, (2) Obese (body fat percentage ≥35%), and (3) Not on hormone therapy	- Lower TNF-α and higher IL-10/TNF-α ratio in the metabolic syndrome group suggestive of inflammation in metabolic syndrome (*p* = 0.005). - IL-6, IL-10, and IL-10/IL-6 ratio, also suggestive of inflammation, were slightly higher in metabolic syndrome group, although not statistically significant.
Klisic et al., 2023 [2]	- Cohort study - 126 participants - Serum samples - Montenegro	Serum endocan levels in postmenopausal women with metabolic syndrome	(1) Postmenopausal women, (2) BMI 29.3 (26.7–31.7) Kg/m^2^, (3) Not on hormone therapy, and (4) No serious health conditions	- Endocan levels were noted to be almost 2.7 times higher in postmenopausal women with metabolic syndrome (*p* < 0.001), suggestive of endothelial dysfunction and inflammation. - CRP was higher in postmenopausal women with metabolic syndrome (*p* < 0.001), suggestive of inflammation.
Stevens et al., 2020 [3]	- Cohort study - 1534 participants - Serum samples - United States	Metabolomic profiles associated with BMI, and waist circumference	(1) Postmenopausal women, and (2) Different ranges of BMI and waist circumference, (3) Cancer free.	- Higher BMI and waist circumference were positively associated with higher levels of C-reactive protein and C-peptide (*p* < 0.001), suggestive of inflammation and insulin resistance, while adiponectin levels were inversely associated (*p* < 0.001), suggesting that greater adiposity is associated with adverse metabolic and inflammatory profiles.
Garrison et al., 2017 [4]	- Cohort study - 924 samples - Plasma - United States	Proteomic analysis, immune dysregulation, and pathway interconnections with obesity	(1) Postmenopausal women, (2) overweight/obese, (3) No other chronic diseases and cancer free	- The following proteins were noted to be significantly different between overweight/obese and normal BMI groups in 1 of the 4 study sample sets: Catalase, Aldehyde dehydrogenase, Apolipoprotein A-I, fibrinogen gamma chain, Parkinson disease protein 7, Coagulation factor V, Glyceraldehyde-3-phosphate dehydrogenase, Vimentin, and Annexin A5. - Significantly dysregulated soluble inflammatory mediator pathways include T cell receptor signaling, PI3K–Akt, prolactin, and AGE–RAGE in diabetic complications, along with leptin, Oncostatin M, IL-2, IL-3, IL-4, IL-5, IL-6, IL-11, EPO receptor, TGF-beta, Kit receptor, interferon type I signaling, and transforming growth factor beta receptor signaling were significantly dysregulated in the overweight/obese group compared to those with normal BMI. - Dysregulated cellular stress pathways involve HIF-1, TNF, FoxO, and phospholipase D signaling, with additional alterations in the AGE/RAGE and Oncostatin M pathways, angiogenesis, nanoparticle-mediated receptor signaling, ATM-dependent DNA damage response, alpha 6 beta 4 signaling, Notch signaling, and regulation of apoptotic and stress responses were significantly dysregulated in the overweight/obese group compared to those with normal BMI. - Metabolic dysregulation pathways include AGE–RAGE in diabetic complications, PI3K–Akt, FoxO signaling, and insulin-related pathways such as insulin receptor signaling, response to insulin stimulus, and cellular response to hormone and peptide hormone stimuli were significantly dysregulated in the overweight/obese group compared to those with normal BMI.
Wong et al., 2008 [5]	- Cohort - 290 participants - Blood - United States	Weight loss and its impact on cytokine levels	(1) Postmenopausal, (2) overweight or obese: waist circumference >80 cm and a body mass index of 25–39.9, (4) blood pressure <140/90 mmHg, (5) a low-density lipoprotein cholesterol level between 100 and 160 mg/dL, (6) no current use of cholesterol-lowering medication, (7) no diagnosis of diabetes or use of diabetic medication	- Lower levels of IL-1 receptor antagonist, IL-6, and C-reactive protein (*p* < 0.05), cytokines involved in inflammation, were reported with weight loss. - Lower levels of other inflammatory markers, such as IL-1α, IL-4, IL-10, Interferon-inducible protein-10, Monocyte chemoattractant protein-1, and Tumor necrosis factor-α, and an increase in IL-8 were reported with weight loss, although not statistically significant.

## Data Availability

Not applicable.

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
