# Peer review of "From Menopause to Molecular Dysregulation: Proteomic Insights into Obesity-Related Pathways—A Narrative Review"

_biomedicines, 2025, doi:10.3390/biomedicines13071558_

Round 1

Reviewer 1 Report

Comments and Suggestions for Authors

This review addresses an important topic and is generally well-written. The structured literature search is appropriate, and the synthesis provides useful insights into obesity-related proteomic changes in postmenopausal women.  

However, I recommend minor revisions to improve clarity and consistency:  

1. The manuscript uses a structured search but is described as a narrative review. Please explain the rationale and limitations of this approach.

2. Briefly comment on the heterogeneity in proteomic methods and its impact.

3. Reduce redundancy and clearly distinguish speculation from evidence, especially regarding estrogen-related mechanisms.

4. Consider including a visual summary (e.g., table or diagram) of the key proteomic markers identified across studies to aid reader comprehension.

Author Response

Overall comment to the reviewer: Thank you for reviewing this manuscript and for your comments.

Comment 1. The manuscript uses a structured search but is described as a narrative review. Please explain the rationale and limitations of this approach.

Response 1: Thank you for your insightful comment. While we employed a structured search strategy, our manuscript is presented as a narrative review rather than a systematic review. This approach was intentional, as our primary aim was to develop a foundational understanding of the topic prior to conducting a full systematic review, which we are considering for future work. We acknowledge that this method has limitations, which we have addressed in the manuscript’s limitations section, please see lines 238-240.

Comment 2. Briefly comment on the heterogeneity in proteomic methods and its impact.

Response 2: Thank you for this valuable comment. We acknowledge the significant heterogeneity in proteomic methods used across the included studies. Although all studies included used serum or plasma samples, each investigator chose their own cytokine and proteomic panel likely because a uniformed proteomic profile for obesity in menopause does not exist. The methodological differences can absolutely lead to variability in the results of the studies. This also highlights the importance of a standardized proteomic approach for future research of obesity in menopause. We have revised and added to lines 236-239 to address this comment.

Comment 3. Reduce redundancy and clearly distinguish speculation from evidence, especially regarding estrogen-related mechanisms.

Response 3: Thank you for this important comment. Currently, there is limited direct evidence linking estrogen deficiency to our findings. However, physiological changes during the menopause transition raise the possibility that estrogen—or other declining ovarian hormones—may play a contributory role. While this remains speculative, we have included it as a plausible explanation, supported by appropriate citations, to aid interpretation of the findings. We have also clarified this point in the revised discussion and incorporated the potential influence of other ovarian hormones in response to a related reviewer comment.

Comment 4. Consider including a visual summary (e.g., table or diagram) of the key proteomic markers identified across studies to aid reader comprehension.

Response 4: Thank you for this helpful suggestion. We agree that a visual summary could enhance clarity. However, due to the significant knowledge gaps stemming from limited data currently known, creating a cohesive and representative visual summary proved challenging. We have instead provided a detailed narrative synthesis in our summary table.

Reviewer 2 Report

Comments and Suggestions for Authors

In this study the authors characterized proteomic signatures of obesity and metabolic syndrome in postmenopausal women. They selected 5 cohort studies and found significant heterogeneity in the proteomic targets without clear identification of the role of estrogen deficiency. Overall, the authors concluded that proteomic evidence could attenuate to identification of the high-risk group in this population. The manuscript is relevant and appropriate to the literature presented, including identifying gaps in knowledge, interpreting findings and the significance of other recent research on the topic, and providing a readable conclusion. However, I would like to make several comments.

  1. The authors should clearly identify what articles and records they selected (English written, others) and how they evaluated their quality and representability.
  2. Schema / figure is likely to be reported to clearly explain what gaps of knowledge require to be solved.
  3. It remaines unclear whether other componens of sex hormon profile in postmenopausal women could relate to inflammatory cytokine net alteration

Author Response

Overall response to the author: We thank the reviewer for their overall comments. The reviewer is correct in that based on the five cohort studies we could not identify a direct relationship with estrogen deficiency, although we eluded to the possibility of estrogen deficiency playing a role in these physiological and molecular dysfunctions in our discussion. This is a knowledge gap that requires further dedicated studies.

Comment 1: The authors should clearly identify what articles and records they selected (English written, others) and how they evaluated their quality and representability.

Response 1: Thank you for this comment. In response to the reviewer, we included English only, adults, and human studies. These details are outlined in the methods section, please see lines 103-105. Since this was a narrative and not a systematic review, we did not do a formal risk of bias assessment and quality review, that is only required for systematic reviews.

Comment 2: Schema / figure is likely to be reported to clearly explain what gaps of knowledge require to be solved.

Response 2: Thank you for your comment. We would appreciate clarification regarding your suggestion. If you are proposing the inclusion of a schematic figure to illustrate the existing knowledge gaps, we acknowledge the potential value of such a visual. However, given the limited understanding currently available in this field, defining these gaps comprehensively would be quite challenging. Nonetheless, we have addressed the key unknowns and limitations in our discussion to the extent possible.

Comment 3: It remaines unclear whether other componens of sex hormon profile in postmenopausal women could relate to inflammatory cytokine net alteration

Response 3: Thank you for this comment, you raised a very important point that we may have missed in our discussion. Yes, it is possible that other components of sex hormone profile may be related to the inflammatory cytokine alterations noted in peri and post-menopausal. We have added a sentence to the discussion to highlight this point, please see lines 220-224.  

Reviewer 3 Report

Comments and Suggestions for Authors

I enjoyed reading this manuscript. It is a review about current evidence on proteomic alterations linked to overweight and obesity in peri- and postmenopausal women. The manuscript is well written and has few grammatical issues. I only have some questions and suggestions to improve the quality of the manuscript.

  • Page 4, Table 1 needs an appropriate caption.
  • Page 7, line 197: sig-nature.
  • Page 7, line 197: im-ply.
  • Page 7, line 223: I especially liked that the authors included this paragraph describing the limitations of this study.

Author Response

Overall response to reviewer 3: We are grateful to the reviewer for their comments and the recognition of this manuscript.

Comment 1: Page 4, Table 1 needs an appropriate caption.

Response 1: Thank you for pointing out this important omission, table 1 now has an appropriate caption. Please see line 148.

Comment 2: Page 7, line 197: sig-nature.

Response 2: This correction has been made, please see line 197.

Comment 3: Page 7, line 197: im-ply.

Response 3: This correction has been made, please see line 217 not 197.

Comment 4: Page 7, line 223: I especially liked that the authors included this paragraph describing the limitations of this study.

Response 4: Thank you for your comment.